# Biomarkers of Brain Dysfunction in Perinatal Iron Deficiency

**DOI:** 10.3390/nu16071092

**Published:** 2024-04-08

**Authors:** Raghavendra B. Rao

**Affiliations:** 1Division of Neonatology, Department of Pediatrics, University of Minnesota Medical School, Minneapolis, MN 55455, USA; raghurao@umn.edu; 2Masonic Institute for the Developing Brain, Minneapolis, MN 55414, USA

**Keywords:** biomarker, brain, ferritin, hemoglobin, iron, iron deficiency, neurodevelopment, perinatal, reticulocyte hemoglobin, zinc protoporphyrin-to-heme ratio

## Abstract

Iron deficiency in the fetal and neonatal period (perinatal iron deficiency) bodes poorly for neurodevelopment. Given its common occurrence and the negative impact on brain development, a screening and treatment strategy that is focused on optimizing brain development in perinatal iron deficiency is necessary. Pediatric societies currently recommend a universal iron supplementation strategy for full-term and preterm infants that does not consider individual variation in body iron status and thus could lead to undertreatment or overtreatment. Moreover, the focus is on hematological normalcy and not optimal brain development. Several serum iron indices and hematological parameters in the perinatal period are associated with a risk of abnormal neurodevelopment, suggesting their potential use as biomarkers for screening and monitoring treatment in infants at risk for perinatal iron deficiency. A biomarker-based screening and treatment strategy that is focused on optimizing brain development will likely improve outcomes in perinatal iron deficiency.

## 1. Introduction

Iron deficiency (ID), the most common micronutrient deficiency in the world, affects 30–60% of pregnancies globally [1,2,3]. In the United States, 40% of pregnant women have evidence of ID in the third trimester with Black, Hispanic, teens, and recent immigrants being at a higher risk [3,4,5,6]. Forty to sixty percent of infants born to mothers with ID anemia have evidence of ID in the fetal and neonatal (perinatal) period [7,8]. Additionally, gestational conditions, such as maternal diabetes, obesity, placental dysfunction, and preterm delivery, predispose the offspring to brain ID [9,10,11,12]. Iron is essential for mitochondrial health, energy production, synaptogenesis, neurotransmission, and myelination. Perinatal ID disrupts these processes and impairs brain development, leading to long-term deficits in attention, recognition memory, executive functioning, and neurodevelopmental and intellectual disorders in adolescence and adulthood [7,8,13,14,15,16,17,18]. Perinatal ID also predisposes to early ID in infancy [19,20], a period when brain development is still active, further compounding the adverse effects. Given the risk of long-term neurological deficits, early diagnosis and prompt treatment are necessary for ensuring normal neurodevelopment in perinatal ID.

## 2. Perinatal Iron Metabolism

The fetus is dependent on the mother for iron needs. The regulation of maternal–placental–fetal iron transport is beyond the scope of this review. Excellent reviews are available elsewhere [21,22,23]. Maternal–fetal iron transport occurs throughout gestation. However, 80% of fetal iron accretion occurs in the third trimester, when daily iron delivery approaches 1–2 mg/kg. The total body iron content of a fetus in the third trimester is 75 mg/kg [24]. Approximately 75–80% is in red blood cells (RBC) as hemoglobin, 10% in iron-containing proteins (e.g., cytochromes and myoglobin), and the remaining 10–15% in storage form, primarily as ferritin [24,25].

## 3. Interorgan Prioritization of Iron

Iron is prioritized to RBC over all other organs in a predetermined order during negative iron balance. Storage and tissue iron are depleted first with the liver and skeletal muscle becoming iron deficient prior to the heart and brain [9]. The final competition for available iron is between RBC for heme synthesis and the brain, with the brain becoming iron deficient prior to the onset of anemia [26]. It is brain ID that is responsible for the adverse neurological effects in perinatal ID [27,28,29]. A similar prioritization favoring RBC over other organs occurs during iron repletion, leaving the brain iron deficient even after the resolution of anemia [30]. The efficacy of iron treatment for correcting brain ID and preventing adverse neurological effects is time sensitive [29,31], and iron transport across the blood-brain barrier is developmentally regulated [32,33]. Thus, there is a narrow therapeutic window for correcting brain ID and preventing neurodevelopmental deficits in perinatal ID. Figure 1 depicts the common causes of perinatal ID, the impact of interorgan prioritization on the developing organ systems, and laboratory indicators reflective of those effects.

## 4. Effects of Perinatal ID on Neurodevelopment

Human data and animal models demonstrate that the hippocampus, a brain region central to recognition or explicit memory, and the striatum, important for implicit memory, are highly vulnerable in perinatal ID [27,34,35,36,37,38,39]. In human infants and mouse, rat, and piglet models, perinatal ID leads to a smaller hippocampus and recognition memory deficits that persist long term despite the resolution of ID [27,34,39,40,41,42,43,44,45,46,47,48,49]. Two transgenic mouse models of hippocampal neuron-specific ID confirm that the adverse effects are due to hippocampal neuronal ID and are independent of anemia [29,50,51,52]. Additional studies in full-term infants have demonstrated that perinatal ID is associated with negative emotionality, lower alertness, and soothability in the neonatal period [53]; impaired recognition memory and locomotion in infancy [40,54]; and poor mental, psychomotor, cognitive, and behavioral deficits in childhood [41,55]. Perinatal ID due to maternal gestational diabetes is associated with impaired recognition memory at birth [27] and behavioral abnormalities at 5 years of age [56]. In preterm infants, perinatal ID is associated with increased abnormal reflexes indicative of a poor neurobehavioral status at 37 weeks postmenstrual age (PMA) [57], and increased risk of behavioral abnormalities at 7 years of age [42]. In both full-term and preterm infants, perinatal ID is associated with abnormal auditory brain stem responses (ABR) in the neonatal period that are indicative of delayed myelination [58,59].

## 5. Biomarkers of Brain Dysfunction in Perinatal ID

Given the risk of long-term neurological impairments and a narrow therapeutic window, biomarkers that predict the risk of brain dysfunction early, when it is still possible to reverse it with iron treatment, are necessary to ensure optimal brain development in perinatal ID [60]. A biomarker is defined as a molecular, histologic, radiographic, or physiologic characteristic that is measured as an indicator of normal biological processes, pathogenic processes, or biological responses to an exposure or intervention, including therapeutic interventions [61]. To be of practical use, such biomarkers should be present in an easily accessible compartment (e.g., in blood), not require large specimen volumes or elaborate collection or analytical procedures, and easily determined in a clinical laboratory with immediate availability of results. Such a biomarker for the prevention of brain dysfunction in perinatal ID does not exist, however, and is a major barrier to optimizing brain iron status in neonates [62]. Some electrophysiological measures (e.g., ABR and event-related potentials [ERP]) are sensitive for the early detection of the effects of perinatal ID on the brain [27,40,58,59,63], but either lack specificity (as in the case of ABR) or require expertise for administration and data interpretation (as in the case of ERP). Likewise, magnetic resonance imaging is useful for determining the effects of ID on the developing brain regions [34,64], but is not practical for routine screening in the clinic.

A hematology panel consisting of serum iron indices and RBC parameters is typically employed for assessing perinatal iron status (reviewed in [60,65,66]). However, they are primarily used for achieving hematological normalcy and lack sensitivity and specificity for brain iron status or brain health [67,68]. Furthermore, most of the studies are based on cord blood values [60,66]. While cord blood assessment has practical utility given the relative ease of collection and avoidance of additional phlebotomy blood loss in the infant, cord blood values by themselves are unlikely to provide complete information about the perinatal iron status as they do not account for the postnatal changes due to physiological reasons (e.g., physiological anemia), pathologic conditions (e.g., cyanotic congenital heart defects), or iatrogenic causes (e.g., phlebotomy blood loss, RBC transfusions, the administration of an erythropoiesis-stimulating agent [ESA] and iron supplementation), all of which affect body iron status. The advantages and disadvantages of some of the commonly used serum iron panel and RBC parameters are given in Table 1 and discussed below.

### 5.1. Serum Iron Panel

Laboratory markers of iron metabolism in serum can be categorized as (1) parameters determining storage and functional iron pool (ferritin, serum iron, total iron-binding capacity, transferrin, and transferrin saturation); (2) proteins regulating iron absorption and release from tissues stores (hepcidin, soluble ferroportin-1, and soluble hemojuvelin); and (3) proteins indicating erythropoietic activity (erythropoietin (EPO), soluble transferrin receptor (sTfR), and erythroferrone [69,70]). Some of the more extensively studied perinatal iron indices are reviewed below.

#### 5.1.1. Serum Ferritin

Serum ferritin (SF) indexes storage iron. Cord blood SF levels increase between 23 and 41 weeks of gestation [11,71]. Normative cord blood SF values are available for full-term and preterm infants [11,66]. A low SF is a reliable biomarker of ID as it is not seen in any other condition. An SF ≤ 75 µg/L in the cord blood or neonatal period is typically considered evidence of ID [8,65,72,73]. Such values are associated with a slower ABR in full-term and preterm infants [58,59]; abnormal neonatal reflexes at 37 weeks PMA in preterm infants [57]; and a poor performance in mental and psychomotor tests at 5 years of age in full-term infants [41,56]. A cord SF < 35 µg/L likely indexes brain ID and is associated with impaired recognition memory at birth in full-term infants of diabetic mothers [27]. A problem with SF is that levels could be increased in inflammatory conditions and after packed RBC transfusions [74,75], making SF a poor predictor of ID under those conditions. Similar to low SF, high cord blood SF (≥188 µg/L) is also associated with impaired mental and psychomotor development at 5 years of age in full-term infants [41], most likely due to inflammation or other confounders. Consistent with the latter possibility, a study failed to find an association between SF > 400 µg/L and neurodevelopmental impairment at 8–12 months of age in 24–32-week gestational age preterm infants after controlling for confounders [76].

Unlike cord blood values, SF in the postnatal period is a poor biomarker of ID, risk of neurodevelopmental deficits, or response to iron treatment, especially in extremely low gestational age neonates (ELGAN; gestational age at birth < 28 weeks) [77,78]. A secondary analysis of the NICHD Darbepoetin Trial in ELGAN (Darbe Trial; NCT03169881) found no relationship between SF and ID (defined as low reticulocyte hemoglobin or low mean corpuscular volume) either in the early (≤27 days after birth) or late (≥28 days) neonatal period [77]. An iron balance study in stable, 30-week gestation preterm infants demonstrated a lack of relationship between SF and enteral iron absorption [79]. In another study of ELGAN, there were no relationships between the minimum, maximum, and median SF values during Neonatal Intensive Care Unit (NICU) stay and Bailey Scales of Infant Development (BSID)-III scores at 24 months corrected age [78]. The correlation between the median or maximum SF values and BSID scores improved when infants with evidence of inflammation were excluded from the analysis, highlighting the low sensitivity of SF for predicting neurodevelopmental impairment in the presence of inflammation.

Urine ferritin correlates with SF [80,81] and offers a non-invasive method for screening for perinatal ID. A urine ferritin < 12 ng/mL corrected for urine creatine and specific gravity has 82% sensitivity and 100% specificity for detecting iron-limited erythropoiesis in neonates at risk for ID, with a positive predictive value of 100% [80]. However, the method may not be feasible in small preterm infants as it requires a relatively large volume of urine, and the assay lacks sensitivity in severe ID [81]. The sensitivity of urine ferritin as a biomarker of brain iron status and health has yet to be determined.

#### 5.1.2. Serum Iron, Iron-Binding Capacity, and Transferrin Saturation

Serum iron, iron-binding capacity, and transferrin saturation are measures of the amount of iron being transported in the plasma. The reference ranges in the cord blood for all three are available for full-term and preterm infants [66,82,83]. 

A systematic review of 51 published studies showed that cord blood serum iron concentrations are higher in preterm infants (46.8 μmol/L; 95% CI: 29.7, 63.8 μmol/L) in comparison to full-term infants (28.4 μmol/L; 95% CI: 26.0, 31.1 μmol/L) [83]. Some other studies have found an opposite effect [84,85,86]. The lower serum iron levels in the preterm infants in those studies persisted throughout infancy despite iron supplementation [84,85,86]. Serum iron concentration remains stable during the first month after birth, followed by a decrease until 6 months, and is maintained at that level until 12 months [83,87]. Lower levels are present in the cord blood of infants with perinatal ID due to maternal diabetes [82]. A similar effect was not present in infants born to mothers with ID anemia, in spite of a greater risk for ID and anemia later in infancy [88,89,90]. A selective iron uptake by the fetus is considered responsible for this effect [85]. Only infants born to mothers with severe ID anemia evince lower serum iron levels in cord blood [91]. Serum iron increases acutely after an RBC transfusion, earlier than serum ferritin, in preterm infants [92,93].

Total iron-binding capacity (TIBC) and unsaturated iron-binding capacity (UIBC) reflect the capacity of transferrin, the major iron transporter in plasma, to bind to iron. UIBC refers to the transferrin available for binding with iron (i.e., the binding capacity of transferrin), while TIBC is the sum of serum iron and UIBC [94]. During ID, there is a relative increase in plasma transferrin, giving rise to an increased UIBC and TIBC. In full-term infants, TIBC increases in infancy with the values at 12 months (60 μmol/L) being approximately double of the values at 2 weeks of age (35 μmol/L) [87]. Compared with full-term infants, preterm infants have a higher TIBC throughout infancy [72,84,85,86,87]. TIBC in the cord blood and neonatal period is increased in perinatal ID [95]. A TIBC > 60 μmol/L is typically considered evidence of ID at 6 months, although it is prudent to rely on lab-specific normative values [96,97]. TIBC decreases following RBC transfusions in preterm infants [92].

Transferrin saturation (TSAT) is the percentage of serum iron that is bound to transferrin and is calculated as serum iron/TIBC × 100 [94]. Gestational age-specific cord blood TSAT reference values from 30 to 42 weeks are available [66]. Mean cord blood TSAT is higher in full-term infants (51.7%; 95% CI: 46.5%, 56.9%) than in preterm infants (36.5%; 95% CI: 0.8%, 72.1%) [83]. The values decrease to 23% (95% CI: 10%, 33%) at 6 months of age and are maintained at this level until 12 months of age [87]. A TSAT < 20% is typically considered evidence of ID in the neonatal period in preterm infants [72]. A lower value (<15% or <10%) is used after 4–6 months of age [87,95,98].

There are no human studies that have investigated the sensitivity of the serum iron panel for predicting the risk of brain dysfunction in perinatal ID. In our experiments in a non-human primate model of infantile ID, all four serum iron indices—serum iron, UIBC, TIBC, and TSAT—predicted the future risk of anemia and ID-induced brain metabolic dysfunction in the preanemic period with comparable predictive accuracy [67,99].

#### 5.1.3. Hepcidin

Hepcidin is the central regulator of iron absorption, cycling, and storage in the body. Hepcidin is downregulated during ID, thereby promoting iron absorption in the gastrointestinal tract. Hepcidin-based iron regulation is active in newborn infants [100,101,102,103]. Reference ranges for cord blood hepcidin are available from 24 to 42 weeks of gestation [104]. Compared with preterm infants, cord blood hepcidin levels are higher in full-term infants [83] and the levels double in the first month after birth. A hepcidin level < 16 ng/mL at 4 months of age indicates ID [102]. As with SF, it is possible to determine hepcidin in urine. The urine hepcidin level correlates with the serum hepcidin level in preterm infants [105] and offers a non-invasive screening method. The urine hepcidin/creatinine ratio correlates positively with SF and negatively with the zinc protoporphyrin-to-heme ratio in ELGAN [100]. Hepcidin is affected by RBC transfusions, ESA administration, iron treatment, infection, and inflammation [100,101,104,105,106,107]. There are no data on hepcidin’s sensitivity as a biomarker of brain dysfunction in perinatal ID.

#### 5.1.4. Soluble Transferrin Receptor

Plasma transferrin receptor (TfR) levels increase during infancy, paralleling normal erythropoiesis and without indicating ID [108]. The soluble transferrin receptor (sTfR) is a cleaved fragment of the transmembrane TfR that is derived primarily from reticulocytes [65,66]. sTfR is a marker of intracellular iron status and is increased in tissue ID and iron-deficient erythropoiesis. sTfR is not affected by inflammation. The ratio of sTfR to SF gives the total body iron status and is useful for monitoring responses to iron treatment [109]. A recent study demonstrated an association between an increased sTfR at 5 months of age and poor cognitive function at 5 months and 5 years, suggesting sTfR’s biomarker potential for predicting long-term neurodevelopmental impairments [110]. Similar data for sTfR in the perinatal period are not available.

### 5.2. RBC Parameters

#### 5.2.1. Hemoglobin

As mentioned, majority of the total body iron is in RBCs as hemoglobin (Hgb). Cord blood Hgb levels increase in the third trimester of gestation [111]. Using Hgb for screening has practical utility as the method is universally available and inexpensive. However, Hgb lacks sensitivity for predicting the risk of ID, impending anemia, brain ID, and ID-induced brain dysfunction in infancy [9,10,67,68,99]. Hgb represents the average of values from RBCs of different ages spanning 90 to 120 days and therefore is not a good indicator of the current iron status. The gestational conditions associated with chronic in utero or postnatal hypoxia (e.g., maternal diabetes, placental dysfunction and cyanotic congenital heart diseases) could be associated with brain tissue ID without affecting Hgb levels (which in severe cases of hypoxia could be even increased) [9,50,51,72,112]. Conversely, a low Hgb could be due to physiological reasons (e.g., physiological anemia) or causes other than ID (e.g., hemoglobinopathies, folate deficiency). A recent study found a lack of association between Hgb values in the first 24 h of birth and 2-year neurodevelopment in preterm infants of <32 weeks of gestation [111], highlighting the low sensitivity of Hgb as a biomarker of infant brain health.

#### 5.2.2. Erythrocyte Zinc Protoporphyrin-to-Heme Ratio

The protoporphyrin ring, precursor to the heme molecule, can be detected in circulating RBC [65]. Under conditions of ID, iron is incorporated into the protoporphyrin ring and only a trace amount of zinc is present in the protoporphyrin ring. During negative iron balance, zinc is incorporated in the place of iron, giving rise to an increased zinc protoporphyrin-to-heme ratio (ZnPP/H) [113,114,115]. Thus, increased ZnPP/H indicates iron-deficient erythropoiesis. ZnPP/H in immature RBC has higher sensitivity for detecting mild ID than whole blood ZnPP/H [116]. Reference ZnPP/H values in cord blood and the neonatal period are available for full-term and preterm infants [66,117]. ZnPP/H decreases during the third trimester and inversely correlates with gestational age [113]. Cord blood ZnPP/H is higher in preterm infants and in infants at risk for perinatal ID due to maternal ID, diabetes, obesity, and intrauterine growth restriction [113,117,118,119,120]. A cord blood ZnPP/H > 118 µM/M predicts poor recognition memory at 2 months in full-term infants [40]. A higher cord blood ZnPP/H is also a predictor of ID at 9 months of age [19]. ZnPP/H decreases during the first 6 weeks after birth in preterm infants, followed by an increase [114,121]. Compared with SF, ZnPP/H is affected less by inflammation and RBC transfusions [75,115]. A head-to-head comparison shows that ZnPP/H has greater sensitivity than SF for predicting neurodevelopmental deficits in ELGAN [40,78]. In one study, a lower ZnPP/H while in the NICU was associated with higher mean BSID-III scores in all three (cognitive, language, and motor) domains at 24 months corrected age in ELGAN [78]. The results were not affected when infants with documented inflammation were excluded [78]. A problem with ZnPP/H as a biomarker is that the levels could be affected by RBC transfusions, ESA administration, and iron treatment [113,115,122].

#### 5.2.3. Reticulocyte Hemoglobin Content

Reticulocyte hemoglobin content (Ret-Hgb, typically abbreviated as CHr or RET-H*e*, depending on the analyzer used for determination) reflects hemoglobinization in reticulocytes [123]. Ret-Hgb provides a more real-time information on bone marrow iron status than Hgb since reticulocytes remain in the circulation for a short time (about 48 h). A low Ret-Hgb indicates the presence of ID. Animal studies show that Ret-Hgb has comparable predictive accuracy for prognosticating ID and IDA as the serum iron indices [99]. Ret-Hgb is affected less by inflammation, diurnal variation, or diet than serum iron indices [98,124,125,126,127]. Additional advantages of Ret-Hgb over serum iron indices are that it can be determined using a small blood volume from a capillary sample and its lower cost [128,129]. The small coefficient of variation also makes Ret-Hgb useful for monitoring the iron status of individual infants during ESA administration or iron treatment [130].

Reference Ret-Hgb values for the first 90 days of birth are available for infants between 22 and 42 weeks of gestation [131,132]. The 5th to 95th percentile reference interval in neonates is 25–38 pg [74,131,132]. Ret-Hgb levels decline after birth in both preterm infants and full-term infants, followed by a slow increase once iron supplementation begins [133,134]. A Ret-Hgb < 29 pg had 85% sensitivity and 73% specificity for detecting ID at 3–4 months corrected age in one study [135]. Ret-Hgb provides a better indication of perinatal ID than SF when the two parameters are discordant [74]. On the negative, a low Ret-Hgb can be seen in certain hemoglobinopathies, such as α and β thalassemia, and all the analyzers do not provide Ret-Hgb.

### 5.3. Maternal Peripartum Iron Biomarkers and Infant Neurodevelopment

Maternal ID during pregnancy is the most common cause of perinatal ID worldwide [1,2,3]. In addition to causing perinatal ID and affecting iron-dependent developmental processes in the brain, maternal peripartum ID is associated with poor mother–infant interaction due to maternal depression, stress, and lower cognitive functioning, further impacting offspring neurodevelopment [136,137]. In a prospective study of 132 mother-full-term infant dyads from a well-nourished population at a low risk for ID, the associations between maternal peripartum iron status (Hgb, SF, sTfR, sTfR/SF ratio, and plasma iron at 3 months postpartum) and infant cognitive function at 3 months and 9 months were determined using sophisticated electrophysiological tests [63]. A better maternal peripartum iron status was associated with better infant cognitive performance overall. Higher maternal plasma iron was associated with a faster speed of processing and better memory; higher Hgb with better attention and memory; and lower sTfR and higher sTfR/SF ratio with lower neural response variability, all at 9 months [63]. There was a negative association between maternal plasma iron and infant neural response variability, indicating a slower adaptation to stimuli with higher maternal plasma iron.

### 5.4. Biomarkers of Iron-Dependent Brain Health

The above-mentioned biomarkers primarily index iron status in the heme compartment and not brain iron status or brain health. Molecular biomarkers of iron-dependent brain health are needed for optimizing brain development through early detection and treatment. Two recent developments in this area are reviewed below.

In a study involving human newborn infants at risk for perinatal ID due to maternal anemia, diabetes, or obesity, cord blood exosomal contactin-2 and brain-derived neurotrophic factor (BDNF), both of which are important for brain development [34,138], correlated with cord blood SF [12]. Exosomes are small, cell-derived vesicles found in all the biofluids, carry the same classes of molecules as the parent cell, and thus function as a snapshot of their cell of origin. Exosomal contactin-2 levels were lower in male infants at risk for perinatal ID, while BDNF levels were higher in female infants, suggesting their potential use as sex-specific biomarkers of brain health [12]. Lower cord blood BDNF concentration with maternal anemia has been reported previously and was associated with decreased hippocampal volume at birth [34]. The influence of sex was not assessed in that study.

Our research using a well-characterized nonhuman primate model of infantile ID [30,139,140,141,142,143,144] and proteomic and metabolomic analyses of paired serum and cerebrospinal fluid (CSF) samples has discovered serum biomarkers of metabolic brain dysfunction in the preanemic stage of ID [67,99,141,145,146,147,148]. Several neurologically important metabolites (dopamine, serotonin, and *N-*acetyl-aspartyl-glutamate) were present in the sera in the preanemic period and perturbed by parenteral iron treatment [146,147,148]. Additional studies in this model confirmed the presence of many neurologically important proteins and metabolites representing lipid metabolism, precursors of neurotransmitters, purines, and xenobiotic molecules, and acute phase proteins in the serum and CSF in the preanemic period [67,146]. Among these, homostachydrine and stachydrine demonstrated parallel changes in comparable magnitude in the two compartments [67]. Homostachydrine and stachydrine are derivatives of pipecolic acid betaine and proline betaine, respectively. Whereas the biological role of homostachydrine has yet to be completely understood, stachydrine is known to have neuroprotective effects [149]. Lower homostachydrine and stachydrine in ID is consistent with our previous study in this model, which also showed that both metabolites respond to parenteral iron treatment [148]. Given that nonhuman primate infants have a similar trajectory of brain development and metabolic demand as human infants, and the two species have identical metabolites and in similar concentrations in the sera [150,151], these data have translational relevance.

Our recent study also evaluated the sensitivity of conventional iron panel and RBC indices for predicting the risk of metabolic brain dysfunction in this nonhuman primate model [67]. Serum iron indices (TSAT, TIBC, and UIBC) and RET-H*e*, but not Hgb or other RBC indices, predicted the future risk of metabolic brain dysfunction. The predictive accuracy of RET-H*e* was comparable to that of serum iron indices. A RET-H*e* < 30 pg at 2 weeks of age accurately predicted the risk of an abnormal CSF metabolite profile at 4 months of age in all infants [67]. These promising results require corroborating studies in human infants before Ret-Hgb could be recommended for screening and treatment of perinatal ID in clinical practice.

## 6. Biomarker-Based Iron Supplementation for Optimizing Neurodevelopment

Pediatric societies in North America and Europe currently recommend a universal iron supplementation strategy for full-term and preterm infants [152,153,154,155]. These recommendations lack uniformity in iron dose, time of initiation, and duration of supplementation. They also do not consider the iron status of individual infants. Thus, there is a potential for under- or over-treatment. Moreover, the recommendations focus on hematological normalcy and not optimization of brain development. Recent data suggest that a standardized biomarker-based iron dosage strategy in preterm infants addresses some of these limitations and results in a higher cumulative iron dose, fewer transfusions, and potentially better neurodevelopment without increasing morbidities [45,156,157]. A biomarker-based supplementation strategy would also avoid or delay unnecessary iron supplementation in iron-replete infants [156].

Studies reporting a biomarker-based iron supplementation strategy, typically in the context of ESA administration in ELGAN, have used TSAT, SF, or ZnPP/H for determining the timing of supplementation and adjusting the iron dose [45,72,156,157]. Among these, only the study by German et al. has evaluated the effects on neurodevelopment [45]. This secondary analysis of ELGAN enrolled in the Preterm Erythropoietin (Epo) Neuroprotection Trial (the PENUT Trial; NCT01378273) [158] found that a standardized iron supplementation strategy with dosage adjustments based on SF or ZnPP/H at 14 days and 42 days after birth resulted in a higher daily iron delivery at 60 days in the placebo and Epo groups (3.6 mg/kg in the placebo group and 4.8 mg/kg in the Epo group; range, 0 to 14.7 mg/kg; IQR 2.1–5.8 mg/kg) than the dose currently recommended by the pediatric societies (2–3 mg/kg per day) [152,154]. There was a positive association between cumulative iron dose at 60 days and BSID-III cognitive scores at 2 years of age. A higher cumulative iron dose was also associated with better, but statistically not significant, motor and language scores. In all three domains, infants treated with Epo had better scores than those treated with placebo [45]. A similar association between cumulative iron dose at 90 days and BSID scores could not be demonstrated, highlighting the importance of a postnatal age-specific dosage strategy. The recently completed Darbe trial (NCT03169881) and two ongoing trials (Iron Supplementation and Neurodevelopmental Outcome in ELGANs [NCT04691843], and the Darbe Plus IV Iron to Decrease Transfusions While Maintaining Iron Sufficiency in Preterm Infants [DIVI; NCT05340465]), all of which employ a biomarker-based iron dosage strategy and neurodevelopment as the outcome, are expected to provide additional information.

There are limitations for instituting a biomarker-based iron supplementation strategy. A single biomarker is unlikely to be sensitive for predicting both the risk of brain dysfunction and efficacy of treatment. For example, whereas SF may be useful for monitoring for the risk of brain ID (e.g., during ESA therapy) in ELGAN [72], it does not appear to be sensitive to assessing response to iron treatment. A recent study found a negative association between SF and cumulative iron dosage, an opposite effect than expected [77]. It is also probably futile to aim for the normalization of SF during iron treatment since storage iron is the last compartment to get replenished. In a previous study, despite close monitoring for ID during ESA therapy and meticulously adjusting the iron dose to maintain TSAT > 20%, 60% of ELGAN had evidence of ID (SF < 75 µg/L) at the conclusion of ESA therapy [72]. An RBC-based biomarker is probably better for monitoring response to iron therapy since iron is prioritized to RBC for heme synthesis during treatment [77,79,113]. Both ZnPP/H and Ret-Hgb appear to be better biomarkers for this purpose. Both are altered in the preanemic period and respond to iron treatment [77,122,134]. Our recent data in nonhuman primate studies suggest that Ret-Hgb may be superior to ZnPP/H for predicting the risk of metabolic brain dysfunction in infantile ID [67,99]. The small coefficient of variation of Ret-Hgb is also conducive for monitoring response iron treatment in individual patients [130].

## 7. Conclusions

Perinatal ID is common and has negative effects on neurodevelopment. A major barrier to optimizing brain development in perinatal ID is the lack of sentinel biomarkers in an easily accessible compartment that predict the risk of impending brain dysfunction and could be used for deciding the timing, dosage, and duration of iron supplementation [62]. Conventional RBC indices lack sensitivity for the early detection of brain dysfunction [67]. While serum iron indices can predict the risk of brain dysfunction in the preanemic period [67], they are not practical for routine screening. Multiple indices have to be determined for assessing the iron-status in various heme compartments [69,70], which requires a relatively large volume of blood via venipuncture. Moreover, the tests may not be readily available in the clinic setting and the results may not be immediately available. Several iron indices lack gestational-age specific reference values, especially in preterm infants. Many are affected by diurnal variation, diet, inflammation, and analytical variability, leading to a large within-subject variability [65,130,159,160]. There are promising serum indicators of iron-dependent brain health in human neonates and animal models [12,34,67,146], but they require additional studies before they can be recommended for clinical use. Preclinical studies show that Ret-Hgb has comparable predictive accuracy as the serum iron panel for predicting the risk of anemia and brain dysfunction in ID [67,99]. Given that Ret-Hgb is a standard component of the hematology panel in some analyzers and thus does not require additional blood collection, with additional supportive data on improved long-term neurodevelopment with a Ret-Hgb-based screening and supplementation strategy, it could become a cost-effective and phlebotomy-sparing single biomarker in perinatal ID.

## Figures and Tables

**Figure 1 nutrients-16-01092-f001:**
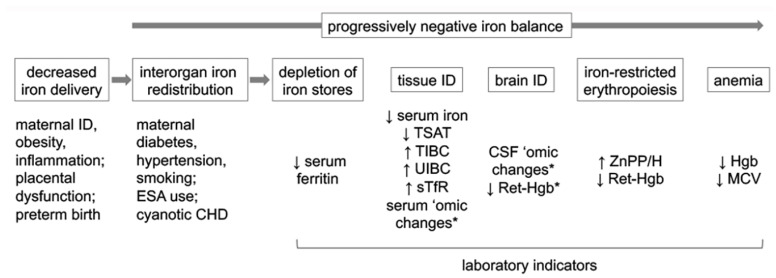
Common causes of perinatal ID, the impact of interorgan prioritization on the developing organ systems, and laboratory indicators reflective of those effects. Abbreviations: CHD, congenital heart disease; CSF, cerebrospinal fluid; ESA, erythropoiesis-stimulating agent; Hgb, hemoglobin; ID, iron deficiency; MCV, mean corpuscular volume; ‘omic, proteomic and metabolomic; Ret-Hgb, reticulocyte hemoglobin content; sTfR, soluble transferrin receptor; TIBC, total iron-binding capacity; TSAT, transferrin saturation; UIBC, unsaturated iron-binding capacity; ZnPP/H, zinc protoporphyrin-to-heme ratio. * based on data from preclinical models.

**Table 1 nutrients-16-01092-t001:** Hematological indices in perinatal iron deficiency.

Laboratory Test	Indicator of	Advantages	Disadvantages
Hemoglobin, hematocrit, mean cellular volume	Anemia	Ease of determination. Immediate availability of results.	Lack sensitivity and specificity for ID, brain ID, and brain dysfunction.
Serum ferritin (SF)	Iron stores	Low SF is specific for ID. Known association between cord SF and short- and long-term neurodevelopment.	Falsely elevated in inflammation. Poor relationship between SF after birth and neurodevelopment.
Serum iron panel (iron, transferrin saturation, total iron-binding capacity, unsaturated iron-binding capacity)	Iron deficiency	Detects preanemic ID. Predicts ID-induced brain dysfunction in the preanemic period *.	Affected by diet and inflammation. Requires additional blood volume. Lack of reference values in preterm infants.
Soluble serum transferrin (sTfR)	Intracellular iron status	sTfR:SF ratio indicates body iron status and useful for monitoring response to iron treatment. Association between postnatal sTfR and neurodevelopment.	No data on relationship between cord/neonatal sTfR and neurodevelopment.
Hepcidin	Regulation of iron absorption	Availability of reference values across the gestational age spectrum.	Altered by multiple factors. No data on sensitivity for detection of brain ID or dysfunction.
Zinc protoporphyrin-to- heme ratio	Iron-deficient erythropoiesis	Predicts impending anemia. Association with neurodevelopment.	Not universally available.
Reticulocyte hemoglobin	Bone marrow iron deficiency	Predicts impending anemia, brain ID, and brain dysfunction *. Component of CBC in some analyzers. Does not require additional blood volume.	Not available in all analyzers. Falsely low in hemoglobinopathies. Sensitivity for predicting long-term neurodevelopment unknown.

CBC, complete blood count; ID, iron deficiency. * based on data from preclinical models.

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
