# Peer review of "Biomarkers of Brain Dysfunction in Perinatal Iron Deficiency"

_nutrients, 2024, doi:10.3390/nu16071092_

Round 1

Reviewer 1 Report (Previous Reviewer 2)

Comments and Suggestions for Authors

An interesting, well-written manuscript on the role of iron in central nervous system dysfunction in prenatal life.

I only have a few minor comments:

1. table 1 - this data is widely known and I would consider removing it or replacing it with a more attractive diagram,

2. I would add a table to the summary showing the pros and cons of specific laboratory markers,

3. in conclusion, the author can discuss whether the latest iron metabolism parameters can be useful in the assessment of iron deficiency. Please use these publications: Cancers (Basel). 2023;15(4):1041; Children (Basel). 2023;10(5):870.

Author Response

Response to Reviewer 1 comments

An interesting, well-written manuscript on the role of iron in central nervous system dysfunction in prenatal life.

Thank you for your kind words, and suggestions.

I only have a few minor comments:

  1. table 1 - this data is widely known and I would consider removing it or replacing it with a more attractive diagram

I have replaced table 1 with a figure as recommended.

  1. I would add a table to the summary showing the pros and cons of specific laboratory markers

I have provided a table showing the pros and cons of different biomarkers.

  1. in conclusion, the author can discuss whether the latest iron metabolism parameters can be useful in the assessment of iron deficiency. Please use these publications: Cancers (Basel). 2023;15(4):1041; Children (Basel). 2023;10(5):870.

I have incorporated the 2 citations recommended by the reviewer in the revised manuscript (citations 69 and 108).

Reviewer 2 Report (New Reviewer)

Comments and Suggestions for Authors

1.       This review describes the most recent information on possible biomarkers for determining infants at risk for ID-induced impairment of neurodevelopment. This is an important, clinically relevant topic that will be interesting to clinicians, developmental neurobiologists, and basic nutrition scientists.

2.       The manuscript is well organized and readable, using helpful subheadings for different possible markers.

3.       One change that could improve the content would be to add a subheading for Tf saturation and TIBC. The manuscript merely mentions these two possible markers in passing near the end of the paper (lines 243-244) but gives no data or details about either of these, just the simple statement that, “Serum iron indices (transferrin saturation [TSAT], total iron binding capacity and unbound iron binding capacity) and RET-He, but not hemoglobin and other RBC indices, predicted the future risk of metabolic brain dysfunction.” RET-He has its own section with an explanation of the strength of the data, as does ZnFF/H, Hepcidin, soluble Tf, ferritin and Hemoglobin, so it is not clear why Tf saturation and TIBC are not fully discussed. Adding this information in their own subheading would strengthen the conclusions of this manuscript.

4.       Another thing the authors might consider is having a more definitive conclusion section. The manuscript clearly lays out which indices are good biomarkers of brain iron and which are not. But the conclusion of this manuscript is rather vague. Consider listing the strongest and most promising markers in the conclusion.

5.       There are a number of minor editorial changes that are needed including:

a.       Reformatting Table 1 to left justify the lines (rather than centering them in the columns) would make the table more readable.  

b.       Line 8 Change to “bodes poor FOR neurodevelopment…”

c.       Line 12 Change to “variation in body iron stores…”

d.       Line 94 “phlebotomy loss”. Change to “phlebotomy loss of blood” or to “blood loss due to phlebotomy.”

e.       Line 160 Results were NOT affected

f.        Line 237 12-fold

g.       Line 238 have a similar trajectory

Comments on the Quality of English Language

My editorial suggestions on word choice are relatively minor and will be easy to correct. However, it is possible that I missed some other minor errors so the manuscript should be checked carefully by an editor for minor errors. 

Author Response

Response to Reviewer 2 comments

This review describes the most recent information on possible biomarkers for determining infants at risk for ID-induced impairment of neurodevelopment. This is an important, clinically relevant topic that will be interesting to clinicians, developmental neurobiologists, and basic nutrition scientists.

Thank you for your kind words, and the suggestions.

  1. The manuscript is well organized and readable, using helpful subheadings for different possible markers.

Thank you.

  1. One change that could improve the content would be to add a subheading for Tf saturation and TIBC. The manuscript merely mentions these two possible markers in passing near the end of the paper (lines 243-244) but gives no data or details about either of these, just the simple statement that, “Serum iron indices (transferrin saturation [TSAT], total iron binding capacity and unbound iron binding capacity) and RET-He, but not hemoglobin and other RBC indices, predicted the future risk of metabolic brain dysfunction.” RET-He has its own section with an explanation of the strength of the data, as does ZnFF/H, Hepcidin, soluble Tf, ferritin and Hemoglobin, so it is not clear why Tf saturation and TIBC are not fully discussed. Adding this information in their own subheading would strengthen the conclusions of this manuscript.

I have added a section on serum iron, TIBC and TSAT.  As the 3 are related (all measure the amount of iron being transported in the plasma) and are conventionally discussed together, I have provided a common section for the 3.

  1. Another thing the authors might consider is having a more definitive conclusion section. The manuscript clearly lays out which indices are good biomarkers of brain iron and which are not. But the conclusion of this manuscript is rather vague. Consider listing the strongest and most promising markers in the conclusion.

I have revised the conclusion section as recommended by this reviewer.

  1. There are a number of minor editorial changes that are needed including:
  2. Reformatting Table 1 to left justify the lines (rather than centering them in the columns) would make the table more readable.  

Thank you for this suggestion. Based on Reviewer 1’s recommendation, I have replaced Table 1 with a figure in the revised manuscript.

  1. Line 8 Change to “bodes poor FOR neurodevelopment…”

Revised as recommended.

  1. Line 12 Change to “variation in body iron stores…”

Revised as recommended.

  1. Line 94 “phlebotomy loss”. Change to “phlebotomy loss of blood” or to “blood loss due to phlebotomy.”

Revised as recommended.

  1. Line 160 Results were NOT affected

Revised as recommended.

  1. Line 237 12-fold

Revised as recommended.

  1. Line 238 have a similar trajectory

Revised as recommended.

Round 2

Reviewer 1 Report (Previous Reviewer 2)

Comments and Suggestions for Authors

The author has addressed all of my previous comments and revised the manuscript.

This manuscript is a resubmission of an earlier submission. The following is a list of the peer review reports and author responses from that submission.

Round 1

Reviewer 1 Report

Comments and Suggestions for Authors

The review by Raghavendra B. Rao elaborates on the importance of biomarkers that can predict brain dysfunction in relation to prenatal iron deficiency. It is well-written and organized and allows the readers to clearly understand and compare the utility of the various biomarkers currently being used or tested. Scientists and clinicians working in this area will find it to be a useful literature resource.

Some comments to address: 

1.     It will be helpful to include a table with the list of the biomarkers, discussing their key features, advantages, and disadvantages along with references

2.     Line 86: please use the values/ range of cord blood Hgb- normal vs increase

3.     Lines 115-116: ‘ZnPP/H ratio from immature erythrocytes has 115 higher sensitivity than whole blood ZnPP/H ratio for detecting mild ID’. Please explain how ZnPP/H ratio in immature erythrocytes vs whole blood is measured, and what accounts for the difference in sensitivity.

4.     Lines 126-127: ‘Urine hepcidin/creatinine ratio cor-126 relates positively with SF and negatively with ZnPP/H ratio in extremely preterm infants.’ Explain in relation to prenatal ID. Is it true for only extremely preterm infants? What about in other conditions of ID?

5.     Line 156: there are several references to ‘compartments’, which are confusing. Please refer to ‘compartments’ in a single context, example with respect to heme vs non-heme throughout the body of the text. Here, I believe it is being referred to Serum vs CSF? Could you use an alternative word to ‘compartment’ in this context?

6.     Minor edits:

(i)             Line 157- correct  ‘thse biomarkers’

(ii)            Line 199- ‘may lead under and over treatment’: ‘to’ is missing

Comments on the Quality of English Language

Please check carefully for spelling and typing errors. 

Reviewer 2 Report

Comments and Suggestions for Authors

An interesting, well-written manuscript on the role of iron in central nervous system dysfunction in prenatal life.

I only have a few minor comments:

1. table 1 - this data is widely known and I would consider removing it or replacing it with a more attractive diagram,

2. I would add a table to the summary showing the pros and cons of specific laboratory markers,

3. in conclusion, the author can discuss whether the latest iron metabolism parameters can be useful in the assessment of iron deficiency. Please use these publications: Cancers (Basel). 2023;15(4):1041; Children (Basel). 2023;10(5):870.